# Antimicrobial Resistance in *Enterococcus* spp. Isolates from Red Foxes (*Vulpes vulpes*) in Latvia

**DOI:** 10.3390/antibiotics13020114

**Published:** 2024-01-24

**Authors:** Margarita Terentjeva, Juris Ķibilds, Jeļena Avsejenko, Aivars Cīrulis, Linda Labecka, Aivars Bērziņš

**Affiliations:** 1Faculty of Veterinary Medicine, Latvia University of Life Sciences and Technologies, LV-3001 Jelgava, Latvia; aivars.berzins@bior.lv; 2Institute of Food Safety, Animal Health and Environment “BIOR”, Lejupes iela 3, LV-1076 Rīga, Latvia; juris.kibilds@bior.lv (J.Ķ.); jelena.avsejenko@bior.lv (J.A.); aivars.cirulis@bior.lv (A.C.); linda.labecka@bior.lv (L.L.); 3Faculty of Biology, University of Latvia, LV-1004 Rīga, Latvia

**Keywords:** *Enterococcus*, *E. faecium*, *E. faecalis*, WGS, cgMLST, *tet*(M), ARGs, AMR

## Abstract

Antimicrobial resistance (AMR) is an emerging public health threat and is one of the One Health priorities for humans, animals, and environmental health. Red foxes (*Vulpes vulpes*) are a widespread predator species with great ecological significance, and they may serve as a sentinel of antimicrobial resistance in the general environment. The present study was carried out to detect antimicrobial resistance, antimicrobial resistance genes, and genetic diversity in faecal isolates of red foxes (*Vulpes vulpes*). In total, 34 *Enterococcus* isolates, including *E. faecium* (n = 17), *E. faecalis* (n = 12), *E. durans* (n = 3), and *E. hirae* (n = 2), were isolated. Antimicrobial resistance to 12 antimicrobial agents was detected with EUVENC panels using the minimum inhibitory concentration (MIC). The presence of antimicrobial resistance genes (ARGs) was determined using whole-genome sequencing (WGS). Resistance to tetracycline (6/34), erythromycin (3/34), ciprofloxacin (2/34), tigecycline (2/34), and daptomycin (2/34) was identified in 44% (15/34) of *Enterococcus* isolates, while all the isolates were found to be susceptible to ampicillin, chloramphenicol, gentamicin, linezolid, teicoplanin, and vancomycin. No multi-resistant *Enterococcus* spp. were detected. A total of 12 ARGs were identified in *Enterococcus* spp., with the presence of at least 1 ARG in every isolate. The identified ARGs encoded resistance to aminoglycosides (*aac(6′)-I*, *ant(6)-Ia*, *aac(6′)-Iih* and *spw*), tetracyclines (*tet(*M), *tet*(L) and *tet*(S)), and macrolide–lincosamide–streptogramin AB (*lnu*(B,G), *lsa*(A,E), and *msr*(C)), and their presence was associated with phenotypical resistance. Core genome multilocus sequence typing (cgMLST) revealed the high diversity of *E. faecalis* and *E. faecium* isolates, even within the same geographical area. The distribution of resistant *Enterococcus* spp. in wild foxes in Latvia highlights the importance of a One Health approach in tackling AMR.

## 1. Introduction

Enterococci are commensal bacteria of the intestines of humans and warm-blooded animals that may enter the general environment, where they may survive for a prolonged time period [1,2]. Enterococci may withstand environmental stressors and inhabit different ecological niches; hence, microorganisms have been isolated from a variety of sources, including aquatic and terrestrial vegetation, fresh and marine waters, sediment and soil, insects and arachnoids, fish, and foods [3,4,5,6,7,8]. Enterococci and *E. coli* are well-recognized faecal pollution indicators and have been isolated from waters contaminated with faecal waste and sewage. The detection of enterococci and *E. coli* has been suggested for the evaluation of drinking water [9]. 

In humans, enterococci are known as opportunistic pathogens, with *E. faecalis* and *E. faecium* being most frequently isolated in cases of hospital-acquired infections, and they are associated with high mortality rates in Europe [10,11]. The treatment of infections caused by *Enterococcus* species is challenging due to their intrinsic resistance to various antimicrobial agents, including cephalosporins, sulphonamides, and aminoglycosides [12]. The ability of *Enterococcus* spp. to acquire additional resistance via, e.g., mobile genetic elements or sporadic chromosomal mutations, raises concerns about the available therapeutical options for the treatment of enterococcal infections [12,13]. Vancomycin-resistant *E. faecium* (VRE) isolates have been recognized as high-priority pathogens by the World Health Organization, for which new antibiotics are urgently needed, and the increasing prevalence of VRE has been reported in hospitals in Europe [14,15]. 

The rise in antimicrobial resistance (AMR) in *Enterococcus* isolates of clinical, animal, and food origins has been widely recognized [16,17]. There is growing evidence that the environment has become an important reservoir of AMR due to contamination from clinical settings and agroecosystems. The One Health approach, which aims to interlink human, animal, and environmental health, recognizes that the transmission of antimicrobial agents, resistant microorganisms, and antimicrobial resistance genes (ARGs) facilitates the spread of AMR, with commensal microorganisms playing an important role [18,19]. Commensal microorganisms—*E. coli* and *Enterococcus* spp.—have been used as indicators of antimicrobial resistance due to their abundance in polluted environments, associations with the intestinal tract, and the easiness of isolation [20]. The ability of these microorganisms to develop and acquire AMR as a result of selective pressure, clonal spread, the dissemination of genetic elements (e.g., plasmids), and co-selection has made them suitable for the long-term monitoring of AMR trends [21]. The detection of AMR in *E. coli* and *Enterococcus* has been established as part of an AMR surveillance programme in humans, animals, and food of animal origin [22].

The environmental dispersion of antimicrobial agents via agri-food systems may facilitate the spread of ARGs, as suggested by the discovery of a close phylogenetic relationship between vancomycin-resistant *E. faecium* (VRE) swine faecal isolates and surface water isolates from Switzerland [23,24]. Despite AMR patterns in *Enterococcus* spp. from clinical and animal production sectors being reported to share sector-specific antibiotic-consumption-related traits, resistant enterococci may serve as vectors for the further dissemination of AMR in the human–animal–environment interface [23,24].

Antimicrobial resistance in *Enterococcus* spp. has been targeted, mainly in productive animals and products of animal origin, to monitor *Enterococcus* spp. antimicrobial resistance trends, but data on the prevalence of resistant enterococci in other animals are largely missing [25,26]. In recent years, wildlife has been reported as a source of multidrug-resistant (MDR) *Enterococcus* spp. [27]. It has been suggested that wildlife could serve as important sentinels of antimicrobial resistance in the environment because their habitats have not been directly exposed to antimicrobial agents, and the presence of AMR-associated indicators such as ARGs has been linked to a spillover from human and agricultural settings [28]. Hence, the occurrence of resistant and MDR strains may indicate the current AMR distribution trends in the environment [29].

Carnivorous species have been reported as being more likely to carry resistant microorganisms [30]. The red fox (*Vulpes vulpes*) is a generalist predator with a wide geographic distribution in Europe, including Latvia [31]. Red foxes are characterized by broad dietary spectra and an ability to adapt to different ecological environments, even in proximity to human habitats. Despite the importance of red foxes to the control of the wildlife population, the role of these foxes in the spread of human and animal pathogens has been recorded [32]. Red foxes have been associated with carrying MDR bacteria due to their foraging behaviour, where they may acquire resistant microorganisms or ARGs from human and animal waste [33]. Since data on antimicrobial resistance in wildlife, including wild carnivores, are missing, the aim of the present study was to detect antimicrobial resistance, resistance genes, and genetic diversity among *Enterococcus* spp. isolates from red foxes (*Vulpes vulpes*).

## 2. Results

### 2.1. Antimicrobial Resistance in Enterococcus Isolates

Altogether, 34 *Enterococcus* isolates were obtained, which were represented by *E. faecium* (n = 17), *E. faecalis* (n = 12), *E. durans* (n = 3), and *E. hirae* (n = 2).

Antimicrobial resistance was identified in 44% (15/34) of the *Enterococcus* spp. using the minimum inhibitory concentration (MIC) detection method. The highest antimicrobial resistance rates were to tetracycline (18%, 6/34), erythromycin (9%, 3/34), ciprofloxacin (6%, 2/34), tigecycline (6%, 2/34), and daptomycin (6%, 2/34). No isolates were resistant to ampicillin, chloramphenicol, gentamicin, linezolid, teicoplanin, or vancomycin. *E. faecium* exhibited resistance to five out of the twelve antimicrobial agents tested, with 10 out the 17 isolates (59%) being resistant to at least one antimicrobial agent. The lowest rates of antimicrobial resistance were recorded for *E. faecalis*, with one out of twelve strains (8%) being resistant to tetracycline (Table 1). 

### 2.2. Distribution of Antimicrobial Resistance Genes (ARGs) in Enterococcus Isolates

A total of 12 ARGs in 32 *Enterococcus* isolates were identified, and ARGs encoding resistance to aminoglycosides, tetracyclines, and macrolide–lincosamide–streptogramin AB were detected. Two out of the thirty-four *Enterococcus* isolates were excluded from further analysis due to low coverage (<30%, 30266 *E. hirae*) or a failure to meet the identification threshold (>90% of target genes, 3802 *E. faecalis*). 

At least one ARG was present in every *Enterococcus* isolate. ARGs were most frequently identified in *E. faecium* (ten ARGs), while only one ARG was found in *E. hirae*. Among the ARGs, the aminoglycoside-resistant determinant *aac(6′)-I* (59%, 20/32) was the most abundant. *tet*(M) (13%, 4/32) and *msr*(C) (53%, 17/32) were the most frequently identified tetracycline- and macrolide-resistant determinants (Table 2). The ARGs exhibited by *E. faecium* were the *ant(6)-Ia* gene responsible for high-level streptomycin resistance (12%, 2/17); the tetracycline-resistance-encoding *tet*(M) (18%, 3/17); *tet*(L) (6%, 1/17); *lsa*(E) (6%, 1/17) of the efflux pump ABC superfamily, conferring resistance to macrolides, lincosamides, and streptogramins A; and lincosamide-resistant genes *lnu*(B) (6%, 1/17) and *lnu*(G) (6%, 1/17). *E. durans* shared only the aminoglycoside-resistant genes *aac(6′)-I* (6%, 1/17) and *aac(6′)-Iih* (6%, 1/17).

All the *Enterococcus* isolates with at least one identified type of phenotypic resistance shared at least one virulence gene, and the number of ARGs in phenotypically resistant isolates ranged from one (three isolates) to eight (one isolate). Phenotypic resistance to quinupristin/dalfopristin was associated with the presence of the *lsa*(A) gene. Phenotypic resistance to tetracycline was confirmed with the presence of *tet*(M) in three *E. faecium* isolates, *tet*(M) and *tet*(L) simultaneously in one *E. faecium* isolate, and *tet*(S) in one *E. faecalis* isolate. High-level tetracycline resistance (MIC ≥ 128) was identified in two *E. faecalis* isolates. The tet genes were not identified in tetracycline-susceptible and tigecycline-resistant isolates (Table 3). Specific genes encoding daptomycin and ciprofloxacin resistance were not identified. In every phenotypically resistant isolate, with the exception of *E. faecalis* (429441), the presence of *aac(6′)-I*, encoding resistance to aminoglycosides, was found. 

### 2.3. Genetic Diversity in Enterococcus Isolates

Since there was a sufficient number of *E. faecalis* and *E. faecium* isolates, genetic diversity was explored using a core genome multilocus sequence typing (cgMLST) approach. For both species, no apparent clustering was observed (Figure 1 and Figure 2). Even within the same geographic region (municipality), the cgMLST genotypes were highly diverse, especially in the case of *E. faecalis*, where over 50% of the cgMLST genes showed different alleles between any of the isolates when compared. However, there were at least some seemingly more closely related isolates among *E. faecium* (isolates 2124 and 429-441 differed in 137/1423 alleles, while the largest difference in this dataset was between isolates 428-644 and 424-211/2, differing in 1251/1423 alleles). Two *E. faecium* isolates from the Tukuma and Ventspils municipalities, which are geographically more distant from the other regions, also appeared to be more genetically distant from the other regions’ isolates (Figure 2). 

## 3. Discussion

*E. faecalis*, *E. faecium*, *E. durans*, and *E. hirae* were the main species isolated, which corresponds to previous findings on the faecal microbiota of red foxes, wild canids, and felids [34,35]. Higher isolation frequencies of *E. faecium* may be related to feeding behaviours, as *E. faecium* was more frequently carried by carnivores [36].

The percentage of identified *Enterococcus* spp. isolates that were resistant to at least one antimicrobial agent was lower than 73% in Portugal [37] and Italy, where 35% of all isolates were susceptible to all the tested antimicrobial agents [38]. In the present study, no MDR isolates were found that resulted in a conflicting finding with those of previous authors, who reported the prevalence of MDR in *Enterococcus* from 3% in red foxes in Portugal [37] to 63% in wild Pampas foxes (*Lycalopex gymnocercus*) in Brazil [35]. Almost every *Enterococcus* isolate from wildlife carnivores shares the MDR patterns identified for wildlife carnivores in Poland [29]. It has been suggested that the proximity of foxes to human habitats, rural colonization, and the urbanization of foxes sharing the same habitat as humans may promote the distribution of antimicrobial resistance and infectious agents [37,39]. In the present study, the samples mostly originated from remote areas with a low population density; thus, limited access to antimicrobial exposure may have reduced the ability of the faecal microbiota to develop antimicrobial resistance [40].

The identified high rates of antimicrobial resistance in *Enterococcus* spp. to tetracycline, erythromycin, and ciprofloxacin were in agreement with the findings of previous research on Carnivora [36,37,38]. Resistance to tetracyclines and ciprofloxacin in *Enterococcus* spp. was found in isolates from wild Pampas foxes, as well as Geoffroy’s cats in Brazil and red foxes in Portugal [34,35]. The AMR patterns of *Enterococcus* spp. isolates from wild animals may resemble domestic animal, food, and clinical isolates [38,41], and they were found to be in line with antimicrobial consumption data from productive animals in Latvia [42]. The antimicrobial resistance of *E. faecium* in calves against quinupristin/dalfopristin has been reported in Latvia [43]. 

Resistance to glycopeptides (vancomycin and daptomycin) and other last-resort antimicrobial agents was not identified in the present study, with the exception of tigecycline resistance in the *E. hirae* and *E. durans* isolates. *Enterococcus* spp. share intrinsic resistance to different classes of antibiotics, including cephalosporins, aminoglycosides, and trimethoprim–sulfamethoxazole [23]. Resistance to first-line antibiotics such as ampicillin and quinolones has been reported in clinical isolates. Acquired resistance to the glycopeptide vancomycin in *E. faecalis* and *E. faecium* significantly limits the choice of antimicrobial agents available for the treatment of vancomycin-resistant *Enterococcus* infections [8]. Oxazolidinones (linezolid), novel tetracyclines (tigecycline), and lipopeptides (daptomycin) may be used for the treatment of vancomycin-resistant *Enterococcus* infections [44]. Therefore, the monitoring of antimicrobial resistance to critically important antimicrobial agents in *E. faecalis* and *E. faecium* is important for tracking the dissemination of antimicrobial resistance in human and animal populations. Despite *E. hirae* and *E. durans* representing a low clinical significance, the identification of tigecycline-resistant *Enterococcus* spp. is concerning, considering the ability of *Enterococcus* to recruit antimicrobial resistance determinants.

Resistance to vancomycin was not identified in the present study. An increasing number of vancomycin-resistant enterococci have been identified in domestic animals and wildlife. In Portugal, 13.5% of the *Enterococcus* spp. isolates from red foxes exhibited phenotypical resistance to vancomycin, as confirmed by the presence of the *van*(A) and *van*(C-1) genes, while in Italy, high-level vancomycin resistance (MIC ≥ 1024 mg/mL) was not associated with the presence of genetic determinants [38,45]. Resistance as high as 41% was identified in wild birds in Italy [46]. A high prevalence of vancomycin-resistant *Enterococcus* spp. in the wildlife in Poland could be explained by possible interactions between different ecosystems [25,47]. 

Almost all of the *Enterococcus* isolates shared ARGs that confer resistance to aminoglycosides. *Enterococcus* spp. intrinsically exbibit low-level resistance to aminoglycosides, with *aac(6′)-I* enzymes being important for amikacin resistance [48]. Our findings on the widespread occurrence of the *aac(6′)* gene in *E. faecium* genomes are in agreement with those of Zaheer et al. [23]. *aac(6′)-Ii*-like genes specific to *E. hirae* and *E. durans*, which confer resistance to the synergy of the association of penicillin and tobramycin, were identified in the present study [49]. The genes conferring resistance to gentamicin include *aph(2′)-Ib*, *aph(2′)-Ic*, *aph(2″)-Id*, and the high-level gentamicin-resistance (HLGR) gene *aac(6′)-Ie-aph(2’’)-Ia* which inactivates all aminoglycosides with the exception of streptomycin. Genes with high-level resistance to streptomycin and kanamycin have not been identified, in contrast to previous studies indicating an overall low spread of ARGs in the environment and wildlife in Latvia [29,34,50,51]. 

*tet*(M) was the predominant gene in our study among tetracycline-resistant isolates, and was among the most frequently identified genes conferring resistance to tetracyclines [52]. *tet*(M) was among the most abundant ARGs in red foxes in Spain and other carnivores—otters (*Lutra lutra*) in Portugal, racoon dogs (*Nyctereutes procyonoides* Gray, 1834), American minks (*Neovison vison* Schreber, 1777), and beech martens (*Martes foina*) in Poland [29,37,50]. Two *E. faecalis* isolates showed high-level resistance to tetracyclines, with a MIC ≥ 128, and one isolate with resistance to tetracycline was associated with the presence of the high-level streptomycin resistance gene *ant(6)-Ia* and the *tet*(M) and *tet*(L) genes. The *tet*(M) gene confers antimicrobial resistance via ribosomal protection, while *tet*(L) mediates the efflux of tetracycline from cells [53]. The *tet*(M) gene was associated with conjugative transposons of the Tn916/Tn1545 family which facilitate the conjugation process via horizontal ARG transfer. Tn916/Tn1545 transposon can promote the transfer of *tet*M, *erm*B, *aph(3″)-IIIa*, or other transposons or plasmids to ensure horizontal gene transfer [54]. Thus, the presence of high-level *Enterococcus* resistance isolates in red foxes may indicate the opportunity for ARG dissemination in wildlife via AMR transfer from highly populated areas to remote locations, with further spread in wild animals and transfer to human habitats. This demonstrates the need to develop a comprehensive understanding of AMR dissemination in wildlife in order to implement a One Health approach for tackling AMR. 

While traditional tetracycline determinants such as *tet*(M) and *tet*(L) have been associated with the increased transcription and expansion of gene copy numbers in tigecycline-resistant phenotypes [55,56], the determinants of resistance to tetracycline in tigecycline-resistant *E. hirae* and *E. durans* isolates were not identified in our study. 

*msr*(C) was most abundant in *E. faecium* isolates (94%), and it is thought to be species-specific. The *msr*A,B,C genes were found in both erythromycin-resistant and -susceptible isolates [38]. The *mrs*(C) gene of *E. faecium* may encode the efflux pump of macrolides [57], and it was identified in all the erythromycin-resistant isolates in the present study. The *lsa*(E) and *msr*(C) genes are reported to be important for lincosamine, streptogramin A, and macrolide resistance, with the *lsa* genes involved in the development of resistance to clindamycin, lincosamides, and dalfopristin [58]. The *lnu*(B) gene confers resistance to lincosamides only, and Nowakiewicz et al. have suggested its circulation between farm animals and wildlife [59]. 

The investigated isolates of *E. faecalis* and *E. faecium* were highly diverse, as assessed using cgMLST typing, and the high diversity of *Enterococcus* isolates from environmental and animal sources has been reported previously [60]. This indicates the absence of the clonal distribution of *Enterococcus* strains among the fox population in the observed geographical region. The genotypes between and within municipalities were highly variable in most cases. The only case indicating the possibility of a common genetic background involved *E. faecium* isolates 428-644 and 426-804, both coming from the Rezekne municipality and displaying resistance against ciprofloxacin. Still, they were separated by a significant 353-allele difference based on their cgMLST genotypes. However, continued monitoring with more dense sampling would be beneficial for assessing the distribution of genetic diversity among resistant *Enterococcus* species in wild carnivore populations. 

The present study reveals the distribution of resistant *Enterococcus* isolates in areas largely untouched by human activities and agricultural practices, which may have an impact on *Enterococcus* antimicrobial resistance rates. This study was limited by the availability of animals from a single geographic area, which is the subject of a national rabies vaccination monitoring program. Further studies including other carnivorous species are intended for the purpose of obtaining a comprehensive overview of the dissemination of antimicrobial resistance in the environment. 

## 4. Materials and Methods

### 4.1. Samples and Sampling

Altogether, 32 dead red foxes (*Vulpes vulpes*) were delivered to the reference laboratory to monitor the efficiency of a peroral rabies vaccination program in wildlife [61,62]. Red foxes were collected from 25 parishes, with one to three foxes per parish (Appendix A). The caecum of each fox was aseptically removed during a necropsy. The external surface of the caecum was disinfected with 70% ethanol and left for 5 min. Then, the caecum was aseptically incised and 1 g of the content of the caecum was collected. 

### 4.2. Microbiological Testing of Samples

An amount of 1 g of the caecal content was transferred into 9 mL of the azide dextrose broth (Biolife, Milan, Italy) and incubated for 24 h at 37 °C. A loop of the enriched suspension was streaked onto Slanetz and Bartley agar (Biolife, Italy) and incubated for 44 ± 4 h at 44 ± 1 °C. After incubation, colonies with a typical red, maroon, or pink colour were subcultured on blood agar (Biolife, Italy) with the addition of 5% defibrinated horse blood (TCS Biosciences Ltd., Buckingham, UK) for 24 h at 37 °C. *Enterococcus* spp. were confirmed using matrix-assisted laser desorption/ionization time-of-flight mass spectrometry (MALDI-TOFF, Bruker, Bremen, Germany). 

### 4.3. Detection of Antimicrobial Resistance in Enterococcus spp. Isolates 

Antimicrobial resistance was detected with the microbroth dilution method using EUVENC test panels (ThermoFisher Scientific, East Grinstead, West Sussex, UK), which are used by the European Union (EU) for the surveillance of AMR in *Enterococcus* spp. The investigated isolates were suspended in saline (0.5 McFarland) and transferred into 11 mL of cation-adjusted Mueller–Hinton (MH) broth (Thermo Fisher Scientific, Landsmeer, Netherlands). The test panels were inoculated with the bacterial suspension in MH broth and incubated at 37 °C for 24 h. In the EUVSEC test panel, the following antimicrobial agents were included (in μg/mL): ampicillin (0.5–64), chloramphenicol (4–128), ciprofloxacin (0.12–16), daptomycin (0.25–32), erythromycin (1–128), gentamicin (8–1024), linezolid (0.5–64), quinupristin/dalfopristin (0.5–64), teicoplanin (0.5–64), tetracycline (1–128), tigecycline (0.03–4), and vancomycin (1–128). The resistance thresholds were determined in accordance with the EUCAST [63]. 

### 4.4. DNA Extraction and Whole-Genome Sequencing (WGS)

Prior to DNA extraction, the bacterial cultures were grown on blood agar (Biolife Italiana, Monza, Italy). For the DNA extraction, the commercially available kit “NucleoSpin Tissue” (Macherey-Nagel, Düren, Germany) was used. Since enterococci are Gram-positive bacteria, a protocol for hard-to-lyse bacteria was used. The protocol included an additional lysozyme step to hydrolyse the peptidoglycan glycosidic bonds in order to obtain better-quality DNA; after that, the standard protocol nr.5 was followed. The purity of the samples was assessed using a NanoDrop spectrophotometer and the quantity was measured using a Qubit fluorometer (both ThermoFisher Scientific, Landsmeer, The Netherlands).

To prepare the libraries for the whole-genome sequencing, an Illumina DNA prep (Illumina, San Diego, CA, USA) library preparation kit was used by following the provided instructions. For a sample quality assessment before pooling, the gel capillary electrophoresis (QIAxcel Advanced Instrument, QIAGEN, Venlo, Limburg, The Netherlands) was used with a high-resolution cartridge and a 50–1500 bp size marker.

The library was sequenced using the Illumina MiSeq next-generation sequencing system (Illumina, San Diego, CA, USA).

### 4.5. Genome Assembly, Detection of Antimicrobial Resistance Genes (ARGs) and Core Genome MLST (cgMLST)

The raw reads were assembled and antimicrobial resistance was identified using an in-house pipeline written in Snakemake (v7.32.3) [64]. First, the quality of the raw reads (and later, the trimmed reads) was assessed with FastQC (v0.12.1) [65]. Then, the raw reads were trimmed using Trimmomatic (v0.39) [66] with the following settings: 2:30:9 LEADING:3 TRAILING:3 SLIDINGWINDOW:4:15 MINLEN:33. The assemblies were created using SPAdes (v3.15.5), with the option of using careful and paired reads as the input [67]. Afterwards, QUAST (v5.2.0) was used to assess the quality of the assemblies [68]. Then, the assemblies were indexed as references and the raw reads were aligned to them using bwa-mem2 (v2.2.1) [69]. The resulting bam files were sorted and indexed with SAMtools (v1.179) [70]. QualiMap (v2.2.2-dev) was used to assess the coverage [71]. Due to the low coverage (<30), two samples were excluded from further analyses (30266 and 427166). The contamination or misidentification of *Enterococcus* species was detected using Kraken 2 (v2.1.3) with the Standard-16 database (v3/14/2023) on the raw reads [72]. MultiQC (v1.14) was used to compile a report on these different statistics to make it easier to assess the different aspects of quality [73]. Finally, on the assemblies, we also ran AMRFinderPlus (3.11.2) [74] with the database version 2023-08-08.2 to detect ARGs (Appendix A). All of the aforementioned computations were performed on the high-performance computing centre cluster at Riga Technical University.

Afterwards, to allow the cgMLST to be determined and to create a minimum spanning tree (MST), the assembled genomes were imported into Ridom SeqSphere+ (v9.0.10) (Ridom, Muenster, Germany) [75]. We determined the cgMLST and created an MST based on the previously existing schemes for *E. faecium* and *E. faecalis* (available at https://cgmlst.org, accessed on 13 November 2023). One sample (3802) did not reach the threshold of >90% target genes, and therefore it was excluded from the MST. The MST was created in GrapeTree (v1.5.0) using the MSTreeV2 algorithm (https://genome.cshlp.org/content/28/9/1395, accessed on 10 December 2023).

## 5. Conclusions

The present study shows the presence of resistant *Enterococcus* species in red foxes in sparsely populated areas without direct anthropogenic effects. The identification of antimicrobial resistance and antimicrobial resistance determinants in *Enterococcus* species isolated from wild carnivores may be the result of environmental contamination and indirect contact with humans and agricultural animals, since wild predators are at the top of the food chain. The presence of resistant enterococci in red foxes highlights the importance of a One Health approach in tackling antimicrobial resistance. 

## Figures and Tables

**Figure 1 antibiotics-13-00114-f001:**
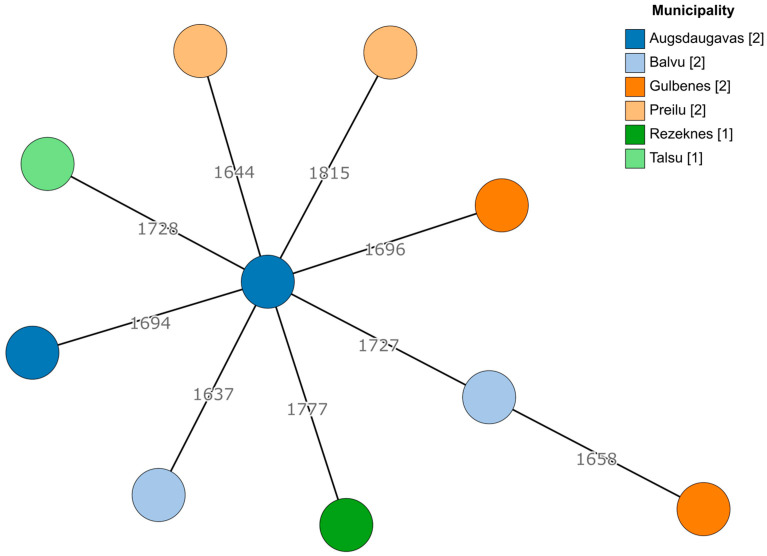
A minimum spanning tree based on the cgMLST scheme of 1972 target genes for *E. faecalis*. Branch lengths are displayed on a logarithmic scale. The number of isolates obtained from each municipality is shown in square brackets.

**Figure 2 antibiotics-13-00114-f002:**
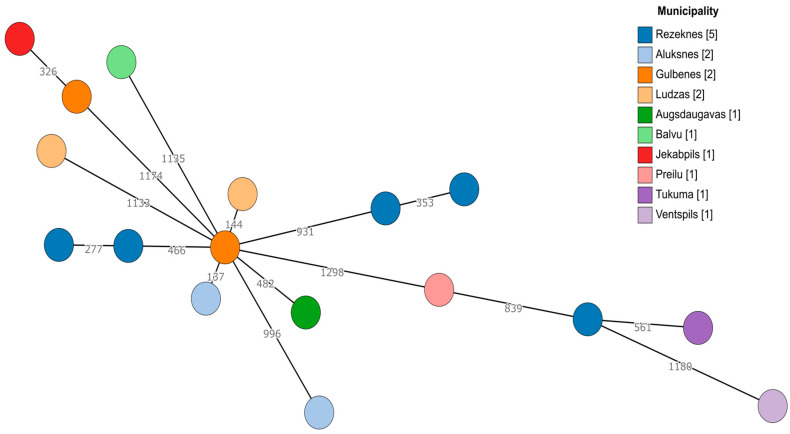
A minimum spanning tree based on the cgMLST scheme of 1423 target genes for *E. faecium*. Branch lengths are displayed on a logarithmic scale. The number of isolates obtained from each municipality is shown in square brackets.

**Table 1 antibiotics-13-00114-t001:** Antimicrobial resistance in *Enterococcus* isolates (n = 34) from foxes.

Antimicrobial Agent	Range of Concentrations (μg/mL)	*E. faecalis* (n = 12)	*E. faecium* (n = 17)	*E. durans* (n = 3)	*E. hirae* (n = 2)	Resistance Threshold (μg/mL)
		No. of Resistant Isolates (%)				
Ampicillin	0.5–64	0 (0)	0 (0)	0 (0)	0 (0)	>4
Chloramphenicol	4–128	0 (0)	0 (0)	0 (0)	0 (0)	>32
Ciprofloxacin	0.12–16	0 (0)	2 (12)	0 (0)	0 (0)	>4
Daptomycin	0.25–32	0 (0)	1 (6)	0 (0)	1 (50)	>4
Erythromycin	1–128	0 (0)	3 (18)	0 (0)	0 (0)	>4
Gentamicin	8–1024	0 (0)	0 (0)	0 (0)	0 (0)	>32
Linezolid	0.5–64	0 (0)	0 (0)	0 (0)	0 (0)	>4
Quinupristin/ dalfopristin	0.5–64	NA	1 (6)	NA	NA	*E. faecalis:* NA, *E. faecium:* >4
Teicoplanin	0.5–64	0 (0)	0 (0)	0 (0)	0 (0)	>2
Tetracycline	1–128	1 (8)	4 (24)	1 (33)	0 (0)	>4
Tigecycline	0.03–4	0 (0)	0 (0)	1 (33)	1 (50)	>0.25
Vancomycin	1–128	0 (0)	0 (0)	0 (0)	0 (0)	>4

NA—not established according to the EUCAST.

**Table 2 antibiotics-13-00114-t002:** Distribution of resistance genes in *Enterococcus* isolates (n = 32) from red foxes.

Enterococcus Species	Aminoglycosides				Tetracyclines			Macrolide–Lincosamide–Streptogramin AB				
	*aac(6′)-I*	*aac(6′)-Iih*	*ant(6)-Ia*	*spw*	*tet*(M)	*tet*(L)	*tet*(S)	*lsa*(A)	*lsa*(E)	*msr*(C)	*lnu*(B)	*lnu*(G)
	No. of Isolates (%)											
*E. faecalis* (n = 12)	1 (8)	0 (0)	0 (0)	0 (0)	0 (0)	0 (0)	1 (8)	11(92)	0 (0)	1 (8)	0 (0)	0 (0)
*E. faecium* (n = 17)	16 (94)	0 (0)	2 (12)	1(6)	3 (18)	1 (6)	0 (0)	1 (6)	1 (6)	16 (94)	1 (6)	1 (6)
*E. durans* (n = 3)	2 (66)	1 (33)	0 (0)	0 (0)	1 (33)	0 (0)	0 (0)	0 (0)	0 (0)	0 (0)	0 (0)	0 (0)
*E. hirae* (n = 2)	1 (50)	0 (0)	0 (0)	0 (0)	0 (0)	0 (0)	0 (0)	0 (0)	0 (0)	0 (0)	0 (0)	0 (0)
Total	20 (63)	1 (3)	2 (6)	1(3)	4 (13)	1 (3)	1 (3)	12(34)	1 (3)	17 (53)	1 (3)	1 (3)

**Table 3 antibiotics-13-00114-t003:** Association of *Enterococcus* spp. resistance phenotypes with the presence of ARGs.

Isolate	Identified Phenotypic Resistance	Identified Antimicrobial Resistance Genes
429441 *E. faecium*	Q/D_8_	*lsa*(A)
429441 *E. faecalis*	TET_64_	*aac(6′)-I*, *tet*(S), *msr*(C)
433771 *E. faecium*	TET_64_	*aac(6′)-I*, *msr*(C), *lnu*(G), *tet*(M)
3808 *E. faecium*	ERY_8_	*aac(6′)-I*, *msr*(C)
426383 *E. faecium*	TET_64_	*aac(6′)-I*, *msr*(C), *tet*(M)
428642/1 *E. faecium*	DAP_8_	*aac(6′)-I*, *msr*(C)
426804 *E. faecium*	CIP_8_	*aac(6′)-I*, *msr*(C)
428644 *E. faecium*	CIP_8_	*aac(6′)-I*, *msr*(C)
427166/2 *E. durans*	TET_64_	*aac(6′)-Iih*, *tet*(M)
23152 *E. faecium*	ERY_8_	*aac(6′)-I*, *msr*(C)
2123 *E. faecium*	TET_128_	*ant(6)-Ia*, *aac(6′)-I*, *lnu*(B), *lsa*(E), *msr*(C), *tet*(L), *tet*(M), *spw*
424211/2 *E. faecium*	TET_>128_, ERY_64_	*aac(6′)-I*, *msr*(C)
424365 *E. hirae*	TIG_0.5_	*aac(6′)-I*
421603/1 *E. durans*	TIG_1_	*aac(6′)-I*

CIP—ciprofloxacin, TET—tetracycline, ERY—erythromycin, DAP—daptomycin, TIG—tigecycline, Q/D—quinupristin/dalfopristin.

## Data Availability

The raw sequence reads from the bacterial isolates included in this study have been deposited in the European Nucleotide Archive under the study accession number PRJEB71100.

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
