# Peer review of "Antimicrobial Resistance in *Enterococcus* spp. Isolates from Red Foxes (*Vulpes vulpes*) in Latvia"

_antibiotics, 2024, doi:10.3390/antibiotics13020114_

Round 1
Reviewer 1 Report
Comments and Suggestions for Authors
Antimicrobial resistance in Enterococcus spp. isolates from red 2 foxes (Vulpes vulpes) in Latvia
Antimicrobial resistance (AMR) is a global health concern that affects both human and animal populations, including wild animals. In recent years, there have been growing concerns about the emergence of antimicrobial resistance in Enterococcus species in various environments, including wildlife. Monitoring antimicrobial resistance in wild animals is essential for a comprehensive approach to addressing this global issue. Wild animals can serve as potential reservoirs for infectious diseases, and the presence of antimicrobial resistance in these populations can complicate disease management and control efforts. The release of antimicrobial agents into the environment, whether through agricultural practices or other human activities, can have long-term effects on ecosystems and contribute to the development of resistance.
The manuscript is well constructed to support this hypothesis that wildlife may serve as a sentinel of AMR. The following are the few suggestions to improve the manuscript:
|
Line # |
Comment |
|
14-15 |
The present study was to detect antimicrobial resistance, antimicrobial …….
The present study was carried out to detect antimicrobial resistance,……
|
|
16 |
E. faecalis (n=12), E. faecium (n=17),…..
Number of isolates may mentioned form higher to lower in number. |
|
16 |
E. hirae (n=2). were obtained from…..
E. hirae (n=2). were isolated from….. |
|
25 |
(tet(M), tet(L), tet(S),
(tet(M), tet(L), tet(S)), |
|
26 |
cgMLST Full form may be written first ….. |
|
27 |
Low prevalence Low prevalence (44%)??? |
|
29 |
AMR may be included among keywords |
|
Introduction |
Some of the references may be updated with recent publications. References from the last few years may be more relatable especially about isolation and detection of AMR in poultry (line# 16-17, line# 62-64). Some of the references are mentioned as under:
Phenotypic and Genotypic Antibiotic Resistance and Virulence Profiling of Enterococcus faecalis Isolated from Poultry at Two Major Districts in Bangladesh
Molecular Characterization of Antibiotic Resistance in Poultry Gut Origin Enterococci and Horizontal Gene Transfer of Antibiotic Resistance to Staphylococcus aureus
|
|
67-68 |
antimicrobial resistance in the environment since they their habitats have not been directly targeted with antimicrobials…..
Please rephrase it. |
|
68 71 165 |
antibiotic-resistant
antimicrobial-resistant? Antibiotic vs antimicrobial: Both terms are used interchangeably. |
|
84 |
Antimicrobial resistance was identified in 44% (15/34) of Enterococcus isolates.
The MIC was used for detection of AMR….. this may be mentioned here. |
|
Table 1 |
NA- not established??? |
|
248 |
incubated for 44 h at 44°C. incubated for 24 h at 44°C.??? |
|
254-263 |
How were the antibiotics chosen for antibiotic susceptibility testing? Were they representative of the common antibiotics used in clinical or agricultural settings? A suitable reference may be added in this regard. |
|
Good Luck! |
|
Comments on the Quality of English Language
Mentioned
Author Response
Reviewer #1
Dear Reviewer,
The authors would like to thank the reviewers for valuable comments and remarks that helped improve the manuscript. Changes in the manuscript are summarized in our response below.
#####
Antimicrobial resistance in Enterococcus spp. isolates from red 2 foxes (Vulpes vulpes) in Latvia
Antimicrobial resistance (AMR) is a global health concern that affects both human and animal populations, including wild animals. In recent years, there have been growing concerns about the emergence of antimicrobial resistance in Enterococcus species in various environments, including wildlife. Monitoring antimicrobial resistance in wild animals is essential for a comprehensive approach to addressing this global issue. Wild animals can serve as potential reservoirs for infectious diseases, and the presence of antimicrobial resistance in these populations can complicate disease management and control efforts. The release of antimicrobial agents into the environment, whether through agricultural practices or other human activities, can have long-term effects on ecosystems and contribute to the development of resistance.
The manuscript is well constructed to support this hypothesis that wildlife may serve as a sentinel of AMR. The following are the few suggestions to improve the manuscript:
Lines 14-15. The present study was to detect antimicrobial resistance, antimicrobial ……..
“carried out” inserted after “was”.
Lines 16. Number of isolates may mentioned form higher to lower in number.
The order of mentioning of Enterococcus species changed in accordance with a number of isolates - E. faecium (n=17), E. faecalis (n=12).
Line 16. “E. hirae (n=2) were obtained”.
“E. hirae (n=2). were obtained from…..” replaced with “E. hirae were isolated”.
Line 25. (tet(M), tet(L), tet(S)).
The parenthesis after tet(S) was inserted.
Line 26. cgMLST. Full form may be written first …..
Full form for cgMLST (a core genome multilocus sequence typing) was inserted.
Line 27. Low prevalence (44%)?
The conclusion was rewritten to highlight importance of spread of AMR in wildlife in accordance to One Health approach.
Line 29. AMR may be included among keywords.
AMR was included in keywords.
Introduction. Some of the references may be updated with recent publications. References from the last few years may be more relatable especially about isolation and detection of AMR in poultry (line# 16-17, line# 62-64).
One of the recommended references was inserted to demonstrate increasing AMR problem in poultry.
Line 67-68. antimicrobial resistance in the environment since they their habitats have not been directly targeted with antimicrobials….. Please rephrase it.
The sentence was rephrased.
Line 68, 71, 165. Antibiotic resistant. Antibiotic vs antimicrobial: Both terms are used interchangeably.
“Antibiotic resistant” was replaced with “resistant” and “AMR”.
Line 84. Antimicrobial resistance was identified in 44% (15/34) of Enterococcus isolates. The MIC was used for detection of AMR….. this may be mentioned here.
“isolates” was deleted, “spp. with MIC detection methods” was inserted.
Table 1. NA-not established.
According to EUCAST, the cut-off values for detection of resistance to quinupristin-dalfopristin are established only for E. faecium.
Line 248. incubated for 44 h at 44°C.
The incubation time and temperature were specified: 44±1°C /44±4 h.
254-263. How were the antibiotics chosen for antibiotic susceptibility testing? Were they representative of the common antibiotics used in clinical or agricultural settings? A suitable reference may be added in this regard.
The EUVENC MIC panels were used in the study. The mentioned panels are recommended in EU for standartized monitoring of AMR in Enterococcus isolated from animals and food for monitoring of AMR trends to critically important antimicrobials ih human medicine.
The reference to EUVENC test panel was inserted.
Sincerely,
Margarita Terentjeva
8 January 2024

Reviewer 2 Report
Comments and Suggestions for Authors
See the attached file

The manuscript needs to be read by an linguistic expert for flow and grammatical errors
Author Response
Reviewer#2
Dear Reviewer,
The authors would like to thank the reviewer for valuable comments and remarks that helped improve the manuscript. Changes in the manuscript are summarized in our response below.
#####
Response to authors
The manuscript written by Terentjeva et al 2023 describes the antibiotic resistance in Enterococcus isolated from red Fox in Lativia region. The manuscript reads well in general and addresses an important issue of public health.
The manuscript has some merit, with following points that needs authors attention to significantly improve its quality
Abstract: is there any ecological significance of red foxes?
Ecological significance of red foxes was added to the abstract and the introduction sections.
Line 14 antimicrobial resistance is suggested to be replaced by “resistant strains” this should be followed through-out the manuscript. The term antibiotic resistance and antibiotic resistance genes are redundant and very confusing.
“Antibiotic-resistant strains” was replaced with “resistant strains” throughout the manuscript.
Line 16 remove full stop after “(n=2)”
The full stop after “(n=2)” was removed.
Line 18 against how many antibiotics
The number of antibiotics (n=12) was added.
Line 22 authors report 13 ARGS while in the text and Table 2 they have mentioned 12 ARGs.
The number of ARGs was corrected, altogether 12 ARGs were detected.
Line 26 write the full form of cgMLST
The full form of cgMLST was added.
How are these findings helpful?
These findings were used for analysis of genetic diversity between Enterococcus isolates to exclude clonal dissemination.
Introduction there are other bacteria such as E. coli, resistant to almost every antibiotic. These closely related enterococci are robust indicators of fecal contamination. Why were these not taken into consideration.
Description of application of commensal E. coli and Enterococcus spp. as fecal and antimicrobial resistance indicators was added to the Introduction.
The authors have targeted wild population of red foxes. Then how do they justify the statement in line 60-61.
In previous studies, red foxes have been suggested as an indicator of environmental contamination with antimicrobials and antimicrobial resistance genes. Therefore, red foxes may indicate the current situation with dissemination of AMR in the general environment.
Results
Line 82-83; which method was used for identification of strains?
Matrix-assisted laser desorption/ionization time-off flight mass spectrometry was used for identification of Enterococcus. Included in the 4.2. Microbiological testing of samples subsection.
Authors are advised to provide colour coded table for all the susceptible, intermediate and resistant Enterococci strains. Was there any standard guideline followed to predict susceptible or resistant profiles.
EUCAST guidelines were used for characterization of susceptible and resistant strains. This information was added to the manuscript. The authors prefer not to change the formatting of the table because it may be not supported by the publisher.
Line 102, 20/24 is equivalent to 83%. Authors are strongly recommended to double check the calculations.
The authors checked and corrected calculations as was suggested by the reviewer.
Line 105, the percentage is missing. Authors should maintain consistency while reporting the results
The results section was checked, percentages, number of isolates and a total number of isolates were added.
Lines 112 to 120 are very confusing and difficult to follow. Authors are advised to paraphrase in consistency with Table 3.
The description of Table 3 was revised with accordance with the content of Table 3.
Line 127 Write the full form of cgMLST. Authors should explain the abbreviations upon its first use.
Full form for cgMLST was inserted.
Which STs were used for phylogenetic comparisons?
STs were used for phylogenetic comparison were added to the supplementary material 1. All isolates of E. faecium and E. faecalis were included regardless of their ST genotype.
Materials and methods, section 4.3 Mention the minimum inhibitory concentrations thresholds and the standards used for categorizing the susceptible/resistant strains
Isolates were categorized into resistant and susceptible according to the EUCAST guidelines. Minimum inhibitory concentration thresholds are shown in Table 1.
Line 291; indexed replace with demultiplexed
Indexing in this context refers to preparation of fasta-formatted sequences for use as a reference against which raw reads will be aligned. Therefore, we did not replace indexing with demultiplexing in this sentence, as it would be incorrect.
Line 291 was rewritten “Then the assemblies were indexed as references and raw reads were aligned to them using bwa-mem2 (v2.2.1)”.
Line 293: so was it 36 isolates that were sequenced or 34. Please clarify the number of samples.
Reason for exclusion of WGS results for 2 isolates was added to the Results, the Materials and Methods and the Supplementary file:
3802 E. faecalis did not reach the threshold of >90% target genes
30266 E. hirae was of low coverage (<30).
In the discussion, authors should specify the threshold of low resistance. Horizontal gene transfer is a mechanism of transferring AMR from remote/wild locations to highly populated human habitats and vice versa. This rationale should be highlighted and elaborated.
In the discussion, analysis of high-level of resistance to tetracycline was included and horizontal gene transfer possibility was highlighted.
Sincerely,
Margarita Terentjeva
8 January, 2024

Reviewer 3 Report
Comments and Suggestions for Authors
Main positive aspects:
AMR represents an important phenomenon with complex implications, less data on wildlife, thus the topic selected by the authors is very relevant
The authors justify very well the study
The bacterial strains AMR profile is obtained using standardized and advanced methods
The manuscript sections are in general well well-constructed and logical
Main aspects that require improvement:
1. Overall, it is an excellent manuscript given the quality of the methodology, and the importance of AMR data in wildlife, still the authors do not use epidemiology to increase the value of their results. As the authors mention the environment and the fecal contamination, as well as the zoonotic risk associated with Enterococcus, One health concept should not be only mentioned, but more detailed applied in the abstract, discussion and conclusion
2. Abstract - the whole text should be modified - too many word repetitions, the authors should add the word "aim" (line 14), give more details on the complex AMR characterization, and rephrase the conclusion to underline the novelty and the importance of the study (add One health)
3. Conclusion: The limitations should be added as part of the discussion section- the authors could mention that further studies extended to the whole country's territory (or maybe other wild species that are also prevalent) would be beneficial/intended etc and add One health
4. Moderate editing of English language is required to remove word repetitions - several words are repeated even within the same sentence or paragraph, the authors should improve this aspect, as well as corrections of the typos (eg feacal)
Comments on the Quality of English LanguageModerate editing of English language required
Author Response
Reviewer#3
Dear Reviewer,
The authors would like to thank the reviewer for valuable comments and remarks that helped improve the manuscript. Changes in the manuscript are summarized in our response below.
#####
Main positive aspects:
AMR represents an important phenomenon with complex implications, less data on wildlife, thus the topic selected by the authors is very relevant
The authors justify very well the study
The bacterial strains AMR profile is obtained using standardized and advanced methods
The manuscript sections are in general well well-constructed and logical
Main aspects that require improvement:
- Overall, it is an excellent manuscript given the quality of the methodology, and the importance of AMR data in wildlife, still the authors do not use epidemiology to increase the value of their results. As the authors mention the environment and the fecal contamination, as well as the zoonotic risk associated with Enterococcus, One health concept should not be only mentioned, but more detailed applied in the abstract, discussion and conclusion.
The authors agree with the reviewer that the manuscript would benefit from epidemiological analysis but to our regret the limited amount of obtained data restricts our possibility to conduct data analysis. The authors intended to expand the number of samples and include the epidemiological aspects in the further study.
The authors highlighted One Health approach importance in the abstract, introduction, discussion and conclusions as it has been recommended by the reviewer.
- Abstract - the whole text should be modified - too many word repetitions, the authors should add the word "aim" (line 14), give more details on the complex AMR characterization, and rephrase the conclusion to underline the novelty and the importance of the study (add One health).
The abstract was revised according to the reviewer suggestions.
- Conclusion: The limitations should be added as part of the discussion section- the authors could mention that further studies extended to the whole country's territory (or maybe other wild species that are also prevalent) would be beneficial/intended etc and add One health
The limitations of the present study were moved to the discussion. Conclusions were modified in accordance with the reviewer’s suggestions to highlight One Health concept.
- Moderate editing of English language is required to remove word repetitions - several words are repeated even within the same sentence or paragraph, the authors should improve this aspect, as well as corrections of the typos (eg feacal)
English editing will be purchased via the MDPI services to proceed further with the manuscript.
Sincerely,
Margarita Terentjeva
8 January, 2024
Round 2
Reviewer 2 Report
Comments and Suggestions for Authors
The authors have addressed most of the comments.
Comments on the Quality of English LanguageMinor spell checks and grammatical errors may be checked.